# Persistent intraocular Ebola virus RNA is associated with severe uveitis in a convalescent rhesus monkey

Gabriella Worwa [1✉], Timothy K. Cooper[1], Steven Yeh[2,3], Jessica G. Shantha[2], Amanda M. W. Hischak [1], Sarah E. Klim [1], Russell Byrum[1], Jonathan R. Kurtz[1], Scott M. Anthony [1], Nina M. Aiosa[1], Danny Ragland[1], Ji Hyun Lee [1], Marisa St. Claire [1], Carl Davis[4], Rafi Ahmed[4], Michael R. Holbrook [1], Jens H. Kuhn [1], Erica Ollmann Saphire [5] & Ian Crozier [1,6✉]

Despite increasing evidence that uveitis is common and consequential in survivors of Ebola virus disease (EVD), the host-pathogen determinants of the clinical phenotype are undefined, including the pathogenetic role of persistent viral antigen, ocular tissue-specific immune responses, and histopathologic characterization. Absent sampling of human intraocular fluids and tissues, these questions might be investigated in animal models of disease; however, challenges intrinsic to the nonhuman primate model and the animal biosafety level 4 setting have historically limited inquiry. In a rhesus monkey survivor of experimental Ebola virus (EBOV) infection, we observed and documented the clinical, virologic, immunologic, and histopathologic features of severe uveitis. Here we show the clinical natural history, resultant ocular pathology, intraocular antigen-specific antibody detection, and persistent intraocular EBOV RNA detected long after clinical resolution. The association of persistent EBOV RNA as a potential driver of severe immunopathology has pathophysiologic implications for under-standing, preventing, and mitigating vision-threatening uveitis in EVD survivors.

[1] Integrated Research Facility at Fort Detrick, National Institute of Allergy and Infectious Diseases, National Institutes of Health; Fort Detrick, Frederick, MD 21702, USA. [2] Emory Eye Center, Emory University, Atlanta, GA 30322, USA. [3] Truhlsen Eye Institute, University of Nebraska Medical Center, Omaha, NE 68105, USA. [4] Emory Vaccine Center, Emory University, Atlanta, GA 30322, USA. [5] Center for Infectious Disease and Vaccine Discovery, La Jolla Institute for Immunology, La Jolla, CA 92065, USA. [6] Clinical Monitoring Research Program Directorate, Frederick National Laboratory for Cancer Research, Frederick, MD 21702, USA. ✉email: gabriella.worwa@nih.gov; ian.crozier@nih.gov

Clinical sequelae in Ebola virus disease (EVD) survivors received urgent clinical, public health, and research attention, for the first time systematically, during and following the 2013–2016 outbreak that occurred in Western Africa. The extent of this outbreak, with 28,652 reported cases and 11,325 deaths, making it the largest recorded, enabled the clinical phenotyping of sequelae in large numbers of EVD survivors. Epidemiologic and clinical characterization newly identified uveitis as particularly common and consequential to EVD survivor's lives. Uveitis associated with EVD had previously only been described in four survivors of a small EVD outbreak in the Democratic Republic of the Congo in 1995[1]. After 2014, large controlled natural-history studies[2,3] and smaller observational patient cohorts[4–8] estimated a high prevalence of uveitis (15–25%) in Western African EVD survivors[2,3]. In survivors from Guinea[5], Liberia[3,7], and Sierra Leone[4,8–10], clinical characterization revealed varied times of uveitis onset (from acute EVD to >1 yr, though typically in the first 3–6 mo of convalescence), locations (anterior, intermediate, or posterior uveitis; pan-uveitis), and severity (mild to vision-threatening). Of these patients with uveitis, many developed secondary ophthalmic complications (e.g., cataract, pupillary and epiretinal membranes, vitreoretinal traction, and retinal detachment) that were vision-threatening and required surgical intervention that was often not available. Albeit only from single observations, case reports on medically evacuated EVD patients[11–13] provided the first high-resolution clinical characterization of EVD-associated uveitis, including, in one patient with severe panuveitis, the detection of high titers of infectious Ebola virus (EBOV) in the aqueous humor more than 2 mo after the resolution of acute EVD and concomitant EBOV clearance from the blood.

How the host-pathogen interaction determines uveitis, especially in the complex environment of ocular immune privilege, remains an open question. The relative contribution of EBOV (or EBOV RNA) persistence versus host immunopathology to ocular inflammation, dysfunction, and damage is unclear. With respect to the virus, the case cited remains the single report of EBOV or EBOV RNA detection in the setting of EVD-associated uveitis. Despite the very high number of EVD survivors with ophthalmic complications, virologic analysis of intraocular fluid sampled during the acute phase of uveitis and early in convalescence has not been attempted or reported from Western Africa. In that population, EBOV RNA has not been detected in aqueous humor sampled during pre-operative evaluations of EVD survivors with post-uveitic cataract[14,15], with the caveat that sampling occurred long after the initial episode of uveitis and in eyes without active inflammation. EBOV-specific immunohistochemistry (IHC) or in situ hybridization (ISH) evaluations of biopsy or post-mortem specimens acquired during the acute phase of uveitis have not been available. Unfortunately, these data serve as the only virologic insight into EVD-associated uveitis, though this is not true of related filoviruses: many decades earlier, a survivor of Marburg virus disease developed uveitis early in convalescence, and infectious Marburg virus was detected in aqueous humor. With respect to host immune responses, immunologic characteristics of EVD-associated uveitis in the periphery, intraocular fluid, and tissue have not been described, including determination of the presence or character of antigen-specific immune responses in the eye. Finally, detailed histopathological characterization of EVD-associated clinical uveitis in humans or other animals has likewise not been reported. Uncertainty around pathophysiology also translates to management in the eye clinic: Though anti-inflammatory management has been appropriately emphasized, the role of antiviral therapeutics in the prevention and treatment of uveitis and its complications remains to be investigated.

Established nonhuman primate (NHP) models used to characterize pathogenesis and evaluate medical countermeasures have not added value for exploration of EVD-associated clinical uveitis outside of post-mortem analysis of rhesus monkey tissues[16]. One challenge is the paucity of survivors in these nearly universally lethal models. Experimental intervention might overcome this barrier, e.g., by successfully treating EBOV-infected NHPs with effective therapeutics. However, prospectively following NHP survivors during extended convalescent periods is associated with ethical concerns; considerable economic, space, and personnel costs in the maximum (animal biosafety level 4 [ABSL-4]) containment setting mandated for EBOV research; and the difficulty of clinically diagnosing and evaluating uveitis in NHPs in that setting.

Here we report the clinical presentation, diagnostic evaluation, and longitudinal follow-up of unilateral uveitis in a rhesus monkey that survived experimental EBOV infection after administration of an EBOV-specific monoclonal antibody (mAb) therapeutic. We present high-resolution multi-dimensional characterization of this intraocular inflammation, including in-life clinical imaging and virologic assessments and, importantly, the immunologic, gross pathologic, and histologic features that reflect the natural history of EVD-associated uveitis absent intervention. Even after apparent clinical improvement, severe ocular pathology was notably associated with the detection of high levels of EBOV-specific antibodies and EBOV RNA in the vitreous fluid of the affected eye. Causality should not be assumed, but the association may implicate persistent EBOV RNA as a possible driver of severe ocular immunopathology and inform future efforts toward understanding, preventing, and treating vision-threatening uveitis in EVD survivors.

## Results

**Acute EBOV infection**. A 5-yr-old female Chinese-origin rhesus monkey (*Macaca mulatta* (Zimmermann, 1780)) was exposed on day 0 with 1000 PFU of EBOV via the intramuscular route, followed by two 25-mg/kg doses (on day 4 and day 7) of human monoclonal anti-EBOV glycoprotein ($GP_{1,2}$) antibody 9.20.1C3 by intravenous infusion[17]. The animal developed a bi-phasic viremia, peaking on day 4 and day 9 post-exposure, as determined by the presence of EBOV glycoprotein (*GP*) nucleic acid (RNA) in serum measured by real-time reverse transcription polymerase chain reaction (RT-qPCR; Fig. 1a, Supplementary Table 1). Between days 4 and 13, the animal clinically scored for hypoactivity and reduced responsiveness (equaling to scores of 1 and 2 on a 4-point euthanasia scale; Fig. 1b, Supplementary Table 1). Decreased platelet counts and elevated aspartate aminotransferase levels, the latter suggestive of hepatic injury and characteristic of acute EBOV infection, were present (Fig. 1c, Supplementary Table 2). Viremia cleared by day 21 coincident with the onset of uveitis. Rhesus-specific anti-EBOV $GP_{1,2}$ IgG antibodies were detected starting on day 12 and remained detectable by enzyme-linked immunosorbent assay (ELISA) until day 99 (Fig. 1a, Supplementary Table 1).

**Ophthalmic presentation and evolution after experimental EBOV infection**. At 21 d after EBOV exposure, eyelid splinting and periorbital erythema of the left eye (OS) were seen during cageside observations and during anesthetized procedures (Fig. 1d). By day 28, OS periorbital swelling and eyelid splinting had improved, but an obvious fibrinous anterior chamber exudate obscured the pupil (cageside pictures from day 29; Fig. 1d). On day 30, close inspection of the OS anterior chamber under anesthesia revealed diffusely dilated iris stromal vessels, posterior synechiae, and expansion of the fibrin plug, but there were no

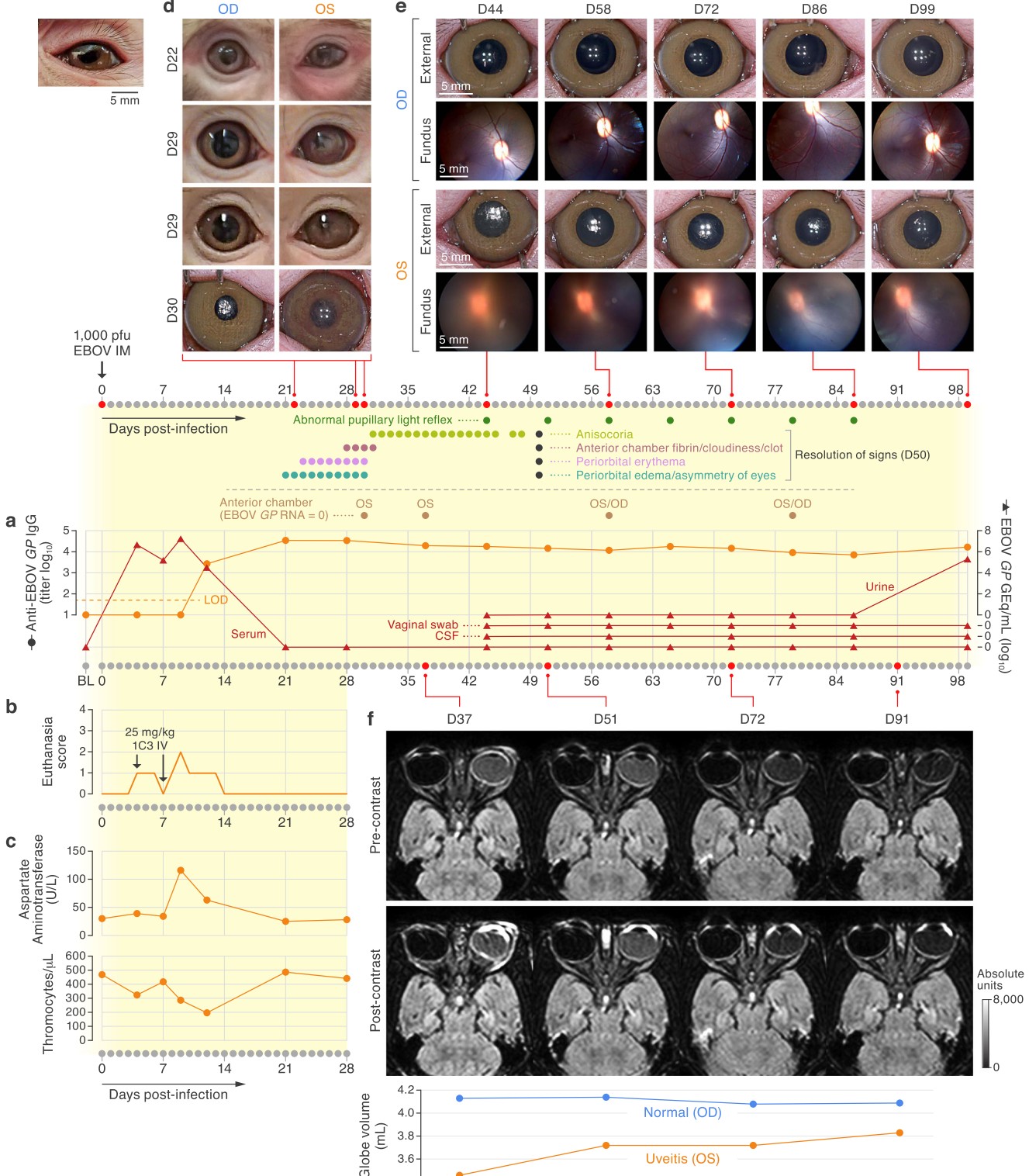

**Fig. 1 Clinical history of acute EBOV infection and uveitis. a** Complete timeline starting with baseline (BL) assessment and ending at necropsy on day 99. Presence of ocular clinical signs and collection of aqueous humor through anterior chamber paracentesis are indicated with dots that are called out with indicator lines. EBOV *GP* genome equivalents (GEq) in sera, determined by RT-qPCR (red triangles), along with rhesus monkey-specific anti-EBOV *GP* IgG antibodies in serum detected by ELISA (orange dots). LOD, limit of detection of assay. **b** Clinical euthanasia scores recorded via cageside observation. Therapeutic human monoclonal antibody 1C3 dosing schedule is indicated with arrows. **c** Concentrations of aspartate aminotransferase (AST) and platelet counts determined in blood are shown over time. **d** Edema, eyelid splinting, anterior chamber fibrin clot, and iris vessel dilation visible from day 21 through day 30. **e** Photographs of the right (OD) and left (OS) eyes, documenting vitreous haze OS with improvement over time. **f** Precontrast and post-contrast magnetic resonance (MR) images of the globes. Volumetric measurements of both eyes extrapolated from MR scans show decreased eye volume OS, suggestive of hypotony. D; day.

obvious corneal precipitates or flare (cloudiness; Fig. 1d). Slit lamp biomicroscopy was not available in ABSL-4 containment, but same-day negative fluorescein staining confirmed intact corneas in both eyes. Obscuration of the pupil gradually disappeared as the fibrinous exudate cleared, but anisocoria and external asymmetry of the eye globes and lids persisted for another 20 d. Direct and consensual pupillary light reflexes were either absent or partial in the left eye until day 86, having presented the only measurable ocular abnormality after day 50 (Fig. 1a, Supplementary Table 3). To evaluate the posterior eye, we began weekly fundus examination with photography on day 37 (Fig. 1e). At the initial evaluation, there was marked OS vitreous haze, obscuring visualization of the fundus and posterior eye; this haze persisted until scheduled euthanasia and necropsy (Fig. 1e). The right eye (OD) was continuously normal (Fig. 1e).

**Magnetic resonance imaging**. Since we were unable to visualize the OS fundus, we performed magnetic resonance imaging (MRI) 37, 51, 72, and 91 d post-exposure (Fig. 1f). The brain was inconspicuous, but gadolinium enhancement of the MRI signal was marked in the anterior chamber and posteriorly near the optic disc, suggesting optic nerve inflammation and papilledema; asymmetric proteinaceous consistency of the vitreous humor indicated vitreous inflammation at all four-time points. Taken together, our observations supported a diagnosis of panuveitis with optic nerve involvement that was most prominent early in the course of uveitis. Because measurement of intraocular pressure using applanation tonometry proved challenging in the ABSL-4 negative-pressure environment, we investigated the volumetric size of the globes by MRI as a structural approximation of decreased intraocular pressure. Recognizing that determining ocular globe volume from MR images is susceptible to measurement error, we utilized the same MRI plane with all anatomical landmarks lined up, encircled the globe, and determined the volume computationally (Supplementary Table 4). At 37, 51, 72, and 91 d post-exposure, the volume of the OS globe measured 3.46, 3.72, 3.73, and 3.83 mL, respectively. In contrast, the OD globe volume fluctuated in the range 4.13–4.08 mL on those days (Fig. 1f). Comparatively lower volumes of the OS globe contour were interpreted as proxy for a relative decrease in intraocular pressure. Overall, the decrease in globe volume and other radiographic abnormalities improved gradually, but enhancement of gadolinium signal was still observed in the OS on day 91.

**Virologic and immunologic assessments**. We collected aqueous humor via anterior chamber paracentesis on day 30 and day 37 from the OS, and on day 58 and day 79 from the OS and unaffected OD. Neither infectious EBOV nor EBOV RNA was detected in aqueous humor in any of the aspirates. However, 5.83 $\log_{10}$ EBOV glycoprotein (GP) genome equivalents (GEq) per mL of vitreous humor OS were detected by RT-qPCR on day 99. This finding was confirmed by a second RT-qPCR targeting the EBOV nucleoprotein (NP). In contrast, EBOV RNA was not detected in unaffected OD vitreous humor on day 99.

To further investigate whether EBOV (or EBOV antigen) was driving intraocular immunologic responses, we quantified anti-EBOV $GP_{1,2}$-specific immunoglobulin G (IgG) and total IgG in the vitreous humor and serum collected terminally to calculate the Goldmann–Witmer coefficient (GWC) as a serologic footprint of intraocular antigen-specific antibody synthesis. A GWC > 4 is considered indicative of a recent intraocular infection even in the absence of detectable pathogen[18]. In contrast to that of the OD (GWC = 0), the calculated GWC of the OS was 5.8, thus confirming intraocular anti-EBOV antibody production

(Fig. 2c). However, infectious EBOV could not be isolated in Vero E6 cell culture, nor could intact virions be identified by electron microscopy of terminal vitreous humor samples.

**Immunopathology**. We further analyzed the cell types and cytokines present in the terminally collected vitreous humor. Cytology of the OD vitreous humor showed only fragments of neuroretina (aspiration artifact, Supplementary Fig. 1), but in the affected OS, vitreous humor cytology qualitatively revealed individual fibroblasts, plasma cells, lymphocytes, and macrophages on a highly proteinaceous background (Supplementary Fig. 2a, b). Out of all live $CD45^+$ cells identified by flow cytometry, 72.43% were $CD3^+/CD8^+$ T-cells (99.65% memory phenotype) and an additional 15.62% were $CD3^+/CD4^+$ T-cells (98.63% memory phenotype; Fig. 2e, Supplementary Table 5, 6 and 7, Supplementary Fig. 3). We further characterized vitreous humor cytokines in both eyes of this NHP and an EBOV-uninfected NHP control (Fig. 2f). CD40 ligand, tumor necrosis factor (TNF), interleukin (IL) 6, IL2, IL23, C-X-C motif chemokine ligand 8 (CXCL8), and vascular endothelial growth factor were uniquely detected in OS vitreous humor but not in the unaffected OD eye. Additionally, interferon gamma, granulocyte colony-stimulating factor, IL2, IL10, IL15, macrophage inflammatory factor 18, and monocyte chemoattractant protein 1 concentrations were elevated in the affected OS compared to vitreous humor from the eyes of the EBOV-naive NHP control (Supplementary Table 8).

**Pathology**. Gross evaluation of the eyes revealed a normal OD (Fig. 2a) but an obviously smaller and grossly abnormal OS globe, with a dense opaque white membrane diffusely present between the vitreous body posteriorly and the iris, lens, and ciliary body anteriorly (Fig. 2a, b). Histopathologically, the OD was normal (Supplementary Fig. 4), but an organizing fibrovascular cyclitic membrane was located between the OS vitreous body posteriorly and the lens, iris, and ciliary body anteriorly (Fig. 2da, db). This membrane was closely adherent to the posterior lens capsule (Fig. 2de). Masson's trichrome staining showed abundant organizing and mature fibrosis within the membrane (Fig. 2da). Two very small foci of cataractous change were noted in the posterior lateral subcapsular lens (Supplementary Fig. 5). There was marked chronic active lymphoplasmacytic inflammation involving the entire inner OS. Moderate numbers of plasma cells and a small number of macrophages and lymphocytes were multifocally present in the stroma of the iris (Fig. 2df) and ciliary body as well as the muscle and trabecular meshwork of the drainage angle, choroid (Fig. 2db, di), retina (Fig. 2dj, dm, dn), and ora serrata (Fig. 2di). Rare Mott cells with Russell bodies (plasma cells with cytoplasm packed with immunoglobulin inclusions, Fig. 2di) and frequent melanin-laden macrophages were present along with scattered pigmentary incontinence (Fig. 2dc, dj). Neuroretina was partially removed in the process of vitreous humor aspiration, but within portions of the retina that were present, mononuclear perivascular cuffing, stromal infiltrates of plasma cells, and angiogenic vessels (connecting to the cyclitic membrane anteriorly) were noted in the inner retina, including the inner nuclear layer (Fig. 2dj, dm and dn). IHC stains confirmed a robust mononuclear cell infiltrate, consisting predominantly of $CD38^+$ plasma cells (Fig. 2dc, dl) and $CD8^+$ T-cells (Fig. 2dd, dk), with a few $CD68^+$ macrophages (Fig. 2dg) and $CD4^+$ T-cells (Fig. 2dh) —a notable difference compared to the cell populations identified in vitreous humor. IHC staining of ocular tissues for EBOV matrix protein (VP40) and $GP_{1,2}$ and ISH staining of vitreous humor cytology and ocular tissues for EBOV genome and antigenome were negative (Supplementary Figs. 6, 7 and 8).

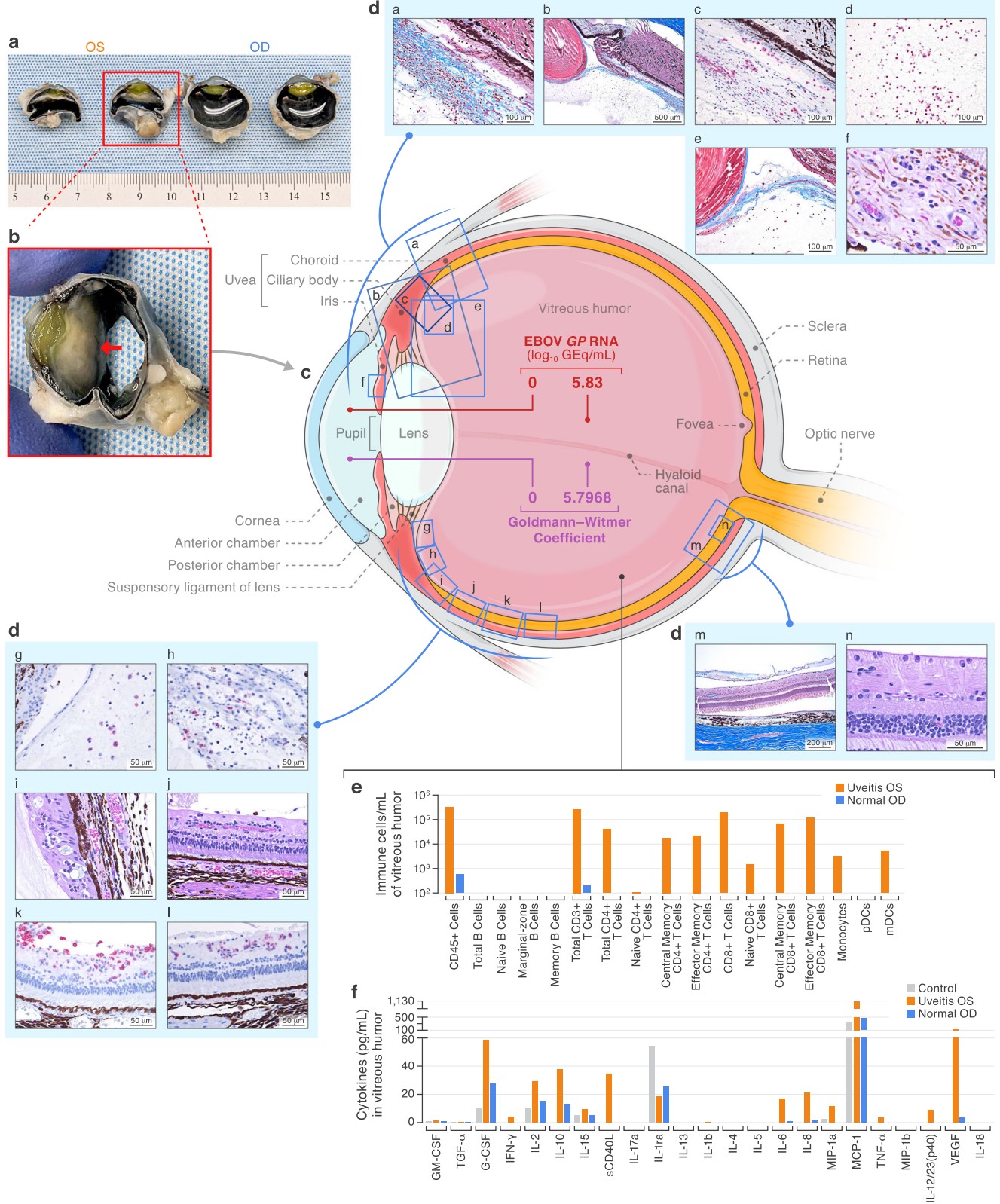

## Discussion

In summary, in a rhesus monkey rescued from experimental EBOV infection by a therapeutic mAb, we observed unilateral anterior uveitis 3–4 weeks after initial infection. Even after clinical improvement in anterior chamber inflammation, clinical progression to panuveitis occurred, with vitreous haze and pupillary dysfunction persisting until 99 d after exposure. MRI confirmed anterior chamber and vitreous inflammation, as well as posterior optic nerve swelling. Though measurement errors are possible when extrapolating the globe volumes from MR images, hypotony was presumably present in the left eye when compared to the right eye. Terminal vitreous humor samples from the affected eye contained inflammatory cytokines and a cellular infiltrate with CD8$^+$ and CD4$^+$ effector memory T-cells,

**Fig. 2 Identification of intraocular EBOV and associated immunopathology.** All samples were collected post-mortem on day 99. **a** Gross pathology of both eyes, bisected, with dense cyclitic membrane present in left eye (OS). **b** Close-up of OS with red arrow pointing at dense membrane. **c** Schematic including quantification of EBOV *GP* genome equivalents (GEq), positive Goldmann–Witmer coefficient, and location of abnormal histopathology of the OS. **d** Abnormal histopathology OS. **da** Masson's trichrome stained section showing a cyclitic membrane with nascent and organizing fibrosis infiltrated by fibroblasts, macrophages, plasma cells, and lymphocytes. **db** Low-power Masson's trichrome stained section demonstrating iridocyclitis, vitritis, and cyclitic membrane. **dc** CD38+ plasma cells infiltrate the anterior retina and cyclitic membrane. **dd** CD8+ T lymphocytes within the vitreous body. **de** Masson's trichrome stained section showing adhesion of the cyclitic membrane to the posterior lens capsule. **df** Plasma cells, macrophages, and lymphocytes infiltrate and expand the iris stroma. **dg** CD68+ macrophages infiltrate the vitreous body. **dh** CD4+ T lymphocytes within the cyclitic membrane. **di** Plasma cells and Mott cells in the ora serrata and anterior choroid. **dj** Macrophages and plasma cells in the retina (primarily ganglion cell and nerve fiber layer) and vitreous body. **dk** CD8+ T lymphocytes infiltrate and expand the inner retina as well as the vitreous body. **dl** CD38+ plasma cells in the inner retina. **dm** Masson's trichrome stain showing macrophages and plasma cells in the posterior retina and vitreous body (detachment is artifactual). **dn** Plasma cells are present in the nerve fiber layer of the posterior retina. **e** Immune cell populations counted by flow cytometry. **f** Proinflammatory cytokine concentrations measured in the vitreous humor OS and OD of this infected animal and from an uninfected control animal.

consistent with viral uveitis[19,20]. The antigen-specificity of the T-cells detected in the vitreous fluid could not be determined. Grossly, the left eye was markedly smaller with a visible opaque cyclitic membrane at the vitreous interface with the ciliary body, lens, and iris. The entire eye—including the uveal tract (iris, ciliary body, and choroid) and portions of the retina—was floridly involved with chronic-active lymphoplasmacytic inflammation, with EBOV-antigen-specific antibody asymmetrically detectable at high levels (GWC > 4), suggesting intraocular EBOV-specific antibody synthesis only in the affected eye. Also in the affected eye, EBOV *GP* and *NP* RNA could be detected in the vitreous humor, but infectivity assays, IHC and ISH evaluation, and electron microscopy findings were negative. We observed the extreme pathophysiologic consequences of intraocular inflammation, including interruption of the visual axis (via a dense cyclitic membrane that almost certainly impaired vision, possibly also by vitreous haze) and hypotony.

These observations may help clarify the relationship between EBOV and the host in the unique environment of ocular immune privilege. Considered a homeostatic mechanism to ensure the preservation of sight in non-renewable ocular tissues, ocular immune privilege is a complex phenomenon inclusive of blood-retinal and blood-aqueous barriers, a relative lack of lymphatics, and an overall immunosuppressive environment[21–23]. EBOV plausibly breaches the eye during the viremia of acute EVD and persists subclinically, at least initially, in the immune-privileged ocular environment. By definition, the appearance of intraocular inflammation, seen three weeks after initial infection in this NHP, implies at least a partial loss of immune privilege in an eye. Viral pathogen breaches of this space may thus begin with subclinical persistence but ultimately lead to severe inflammatory consequences. As immune privilege collapses, persistent EBOV or EBOV RNA might stimulate antigen-specific inflammation that results in ocular tissue dysfunction and damage—and likely also leads to the vision-threatening complications described in human survivors. Prior to more advanced molecular detection of pathogens, the GWC has long been considered evidence of intraocular antigen-specific antibody synthesis, and as such an indirect serologic footprint of a local infection. We were unable to confirm the antigen-specificity of the T-cell intravitreal inflammatory infiltrate, but it is plausible that an EBOV-antigen-specific ocular immune response, especially after the onset of inflammation (the collapse of privilege), would also involve T-lymphocytes. Care is needed to avoid over-interpreting these data or assuming causal relationships from a single observation. However, in the context of vision-threatening intraocular pathology (diffuse uveal lymphoplasmacytic infiltration with fibrosing cyclitic membrane), the detection of high levels of intravitreal EBOV-specific IgG in tissue proximity to detectable EBOV RNA at least query a role for persistent EBOV RNA as a potential driver of uveitis and its complications.

Observations from this rhesus monkey have remarkable similarities to those described in the human EVD survivor with panuveitis and EBOV persistence. With regard to acute EBOV infection, that patient similarly had an extended duration of high-level viremia and received EBOV-specific experimental therapeutic intervention during acute EVD, albeit with convalescent plasma and a small-interfering RNA therapeutic rather than an EBOV-specific mAb. With regard to the clinical syndrome, similar timing (onset in early convalescence), progression (initially anterior to severe panuveitis) and complications that included dense vitritis (impeding the visual axis and leading to near complete loss of vision), hypotony (suggesting ciliary body dysfunction), and optic neuropathy were also noted. For this human survivor, aggressive anti-inflammatory therapy, followed by an experimental anti-viral, plausibly redirected the natural history of severe panuveitis, though it did not prevent longer-term complications that included blinding cataract, requiring surgical intraocular lens replacement, and recurrent mild uveitis managed medically. Notably, over a 2-yr period after the patient's initial improvement, multiple samples were acquired from the aqueous humor and from cataractous lens aspirate; EBOV RNA could not be detected again[13,24,25]. Similarities between these (only) two case reports of clinical uveitis associated with the detection of EBOV RNA in a primate end there; indeed, the pathologic consequences of uveitis in this rhesus monkey survivor's eye illustrate the destructive natural history of this virus–host interaction in the eye when left untreated.

Clarifying the risk factors for uveitis in larger EVD survivor cohorts remains a challenge. In Western African studies, determining risk has been limited by difficulty linking acute to convalescent data in a survivor-specific manner, including details of acute disease severity and viral loads, and the presence or absence of candidate medical countermeasures that could prevent, treat, or cause sequelae. Risk is potentially determined by, or at least informed by, characteristics of acute EVD, including route of exposure, clinical (e.g., ocular-specific symptoms and signs, disease severity), virologic (e.g., peak and duration of viremia), treatment (yes/no, what type), and immunologic features. It is clear that receipt of an EBOV-specific therapeutic was generically related to the NHP's survival (and perhaps to extended duration of a biphasic viremia); what is not clear is any specific relationship between the development of uveitis and receipt of a mAb-based therapeutic in general, or this mAb in particular. The role of pre-existing host genetic or immune factors in the development of uveitis associated with EVD is not clear, including for this rhesus monkey. At the time of necropsy, there was no histopathologic evidence of pre-existing or secondary bacterial, fungal, or other viral infection. Most human infections occur at mucosal interfaces rather than the intramuscular inoculation in our experimental subject; the impact of the route of initial EBOV infection

on the subsequent development of uveitis in human EVD survivors is unclear. In humans and experimentally infected monkeys, however, with few exceptions, the common theme is systemic viremia that affords, especially when at high levels and of extended duration, breach of the blood-retinal and blood-aqueous barriers to seed the eye.

These data have implications for understanding the specific intraocular tissue and cellular pathogenesis of EVD-associated uveitis. The vitreous gel and its local interfaces are of particular interest. A previous study using retrospective examination of tissues from EBOV-exposed NHPs identified EBOV in CD68[+] macrophages particularly located at vitreo-retinal, vitreo-uveal, and vitreo-capsular interfaces in treated and untreated NHPs survivors; clinical uveitis was not noted during the in-life follow-up, but histopathology identified varying incidence of uveitis, vitritis, and retinitis[16]. The histopathologic findings in this rhesus monkey survivor support this vitreous interface as important geography in the development of EVD-associated uveitis and also in the development of vision-threatening ophthalmic complications, i.e., cataract (seen in 6–10% of human EVD survivors), posterior synechiae, pupillary and epiretinal membranes, and vitreoretinal traction (leading to detachment). In this NHP, the dense fibrotic cyclitic membrane proximal to the posterior lens capsule, and obviously impacting the visual axis, suggest this interface is important in development of the fibrinomembranous complications of uveitis; whether the early cataractous change in the posterior lens is related to uveitis cannot be determined but is of interest given the high prevalence of post-uveitic cataract in EVD survivors, including, and unusually, in young children.

With regard to cellular tropism, EBOV-antigen positive CD68[+] macrophages that likely breach the eye during prolonged viremia have been identified[16] but ocular phenotypes in human survivors suggest the involvement or dysfunction of ocular pigmented epithelial cells in the pathogenesis of EVD-associated uveitis. Peripheral chorioretinal scarring is frequently seen in humans and pigmented retinal epithelial (RPE) cells are permissive to EBOV infection in vitro while retaining their immunomodulatory properties[26]. Whether the same is true of other pigmented epithelial cells is not known, and the retina could not be fully assessed in this rhesus monkey. However, ciliary body and iris epithelial cells are also immunomodulatory and plausibly involved in the clinical phenotype. Intraocular pressure measurements and ocular ultrasound imaging performed in human survivors[24] (and proxied in this NHP survivor) suggest ciliary body dysfunction, specifically in the ciliary body epithelial cells that produce aqueous humor. Histopathology in this NHP not only showed lymphoplasmacytic inflammation of the ciliary body; it is likely that the fibrovascular cyclitic membrane is a consequence of ciliary body inflammation[27]. Finally, iris heterochromia observed in the human survivor with panuveitis and EBOV persistence[13,24] is of uncertain pathogenesis but either the direct involvement of pigmented iris epithelial cells, and/or iris stroma atrophy indirectly may be implicated. Indeed, it has been recently demonstrated ex vivo that human iris pigmented epithelial cells are also permissive to EBOV infection, though less susceptible than ex vivo RPEs. Further exploration of the interaction of ocular macrophages, pigmented epithelial cells, ocular fibroblasts, and EBOV ex/in vivo is warranted.

Our findings have implications for the prevention and treatment of uveitis and its complications. Effort is ongoing to establish and enable timely diagnosis and effective management of human uveitis as well as related complications in EVD survivors[14,15,25,28–30]. Thus far, clinical management has focused on early diagnosis to enable important ophthalmic care with anti-inflammatory and cycloplegic targets. EBOV-specific antivirals have been used anecdotally[13] but have not been studied.

Supported by our observations, EBOV-specific therapeutics that can penetrate ocular tissue might be studied for treatment of severe uveitis in EVD survivors, in addition to anti-inflammatory approaches. Given the high prevalence of uveitis in essentially one out of four human survivors, EBOV-specific therapeutics with similar characteristics might be considered for prevention during acute EVD, as well as strategies to maintain blood-ocular barriers and prevent seeding of the eye, e.g., endothelial stabilization.

To the best of our knowledge, we have provided a first report of the detailed clinical features, evolution, and the severe immuno-pathologic consequences of EVD-associated uveitis in a rhesus monkey survivor of experimental EBOV infection. We have reported intraocular EBOV-specific antibody detection and persistent intraocular EBOV RNA months after clearance of viremia and even after clinical resolution. Causality should be not assumed, especially in the context of a single observation, but the association of persistent EBOV RNA as potential driver of severe immunopathology has pathophysiologic implications for understanding, preventing, and treating vision-threatening uveitis in EVD survivors.

## Methods

**Ethics and approvals**. This study used a 5-yr-old female Chinese-origin rhesus monkey (*Macaca mulatta* (Zimmermann, 1780)) sourced through Worldwide Primates. All experimentation was conducted within the biosafety level 4 (BSL-4) facility at the Integrated Research Facility at Frederick (IRF-Frederick), Maryland, USA, and was approved under the National Institutes of Allergy and Infectious Diseases (NIAID), Division of Clinical Research (DCR) Animal Care and Use Committee (ACUC) animal study protocol number IRF-033E. The IRF-Frederick is registered (51-F-0016) with the United States Department of Agriculture (USDA), accredited (777) by the Association for Assessment and Accreditation of Laboratory Animal Care (AAALAC), and registered (D16-00602) by Public Health Service (PHS) Assurance for Laboratory Animal Welfare. Procedures followed the recommendations provided in The Guide for the Care and Use of Laboratory Animals[31] and the American Veterinary Medical Association (AVMA) guidelines for the euthanasia of animals.

**Animal procedures**. High Protein Monkey Diet (No. 5045, LabDiet, St. Louis, MO, USA) was provided daily and fresh water offered ad libitum. After intramuscular (IM) inoculation with 1000 plaque-forming units (PFU) of Ebola virus/H. sapiens-tc/COD/1995/Kikwit-9510621 (EBOV; NR-50306, Lot 9510621, BEI Resources, USA), the monkey was administered 25 mg/kg of human monoclonal antibody (mAb) 9.20.1C3 anti-EBOV glycoprotein (GP) 1 and 2 (GP$_{1,2}$) ("1C3"; Zalgen Labs, Germantown, MD, USA) intravenously (IV) on days 4 and 7 post-exposure. Cageside observations were conducted daily, and euthanasia criteria were assessed based on activity and responsiveness according to a four-point scoring scale (alert [0], slightly subdued [1], withdrawn [2], temporarily recumbent [3], or persistently recumbent [4]). The monkey's eyes were observed and photographed daily at the cageside after noticing onset of uveitis at day 21 post-exposure until resolution of external signs of uveitis at day 50 using an Olympus Tough TG-5 camera (Olympus America Inc., Waltham, MA, USA). The monkey was sedated via IM injection of 15 mg/kg of Ketamine HCl (KetaThesia, Henry Schein, USA) for a baseline assessment prior to exposure and on days 0, 4, 7, 9, 12, 21, 28, 37, 44, 51, 58, 65, 72, 79, 86, 91, and 99 post-exposure for physical examination and collection of venous blood (except day 91) into serum separator and tubes containing K3 ethylenediamine tetraacetic acid (EDTA; Vacuete; Greiner Bio-One, USA). On days 37, 44, 51, 58, 65, 72, 79, 86, 91, and 99 ophthalmic evaluation of both eyes was conducted. The palpebral, corneal, and pupillary light reflexes (direct and consensual) were assessed. Applanation tonometry for the measurement of the intraocular pressure was attempted using a Tono-Pen Avia Vet applanation tonometer (Dan Scott & Associates, Westerville, OH, USA). A 1 % tropicamide solution (Tropicamide Mydriacyl eye drop, Alcon, Fort Worth, TX, USA) was instilled into each eye to elicit mydriasis and facilitate assessment of the fundus. Images of the dilated pupil and the fundus were captured using a Pictor Plus retinal camera (Volk Optical, Mentor, OH, USA). From day 44 to 99, cerebrospinal fluid and vaginal swabs were collected from the monkey while under anesthesia. On day 99, the monkey was euthanized via intravenous overdose of pentobarbital sodium (Fatal Plus Solution; Vortech Pharmaceuticals, Dearborn, MI, USA), followed by a necropsy. Aqueous and vitreous humor were aspirated from the OS and OD and aliquots frozen at −80℃. Fresh vitreous humor was used for a cytology. The volumes were replaced by injecting 10% neutral buffered formalin (NBF) into the posterior chamber followed by fixation of the eyes in 10% NBF.

**Magnetic resonance imaging**. On days 37, 51, 72, and 91 post-exposure, magnetic resonance (MR) imaging of the orbits and brain was performed on a Philips Achieva 3 Tesla clinical magnetic resonance scanner (Philips Healthcare, Cleveland, OH, USA). Gadopentetate dimeglumine agent at 0.1 mL/kg (Bayer Healthcare Pharmaceuticals,

Whippany, NJ 07981, USA) was injected IV prior to each imaging session. After scout scans (done with a large field of view to identify regions for focused scans), the following imaging sequences were acquired with the following parameters:

- Precontrast, 2-dimensional (2D), coronal, short inversion time inversion recovery (STIR): repetition time (TR), 3000 ms; echo time (TE), 60 ms; slice thickness, 2 mm; reconstructed pixel size, 0.49 × 0.49 mm; 22 slices.
- Precontrast T2-weighted (T2W), 2D, axial, turbo spin echo (TSE): TR, 3600 ms; TE, 100 ms; slice thickness, 1.3 mm; reconstructed pixel size, 0.48 × 0.48 mm2; 27 slices.
- Pre- and postcontrast T2W, axial, 2D fluid-attenuated inversion recovery (FLAIR): TR, 10,000 ms; TE, 100 ms; inversion time (TI), 2650 ms; slice thickness, 2 mm; reconstructed pixel size, 0.49 × 0.49 mm; 30 slices.
- Pre- and post-contrast T1-weighted (T1W), 2D, axial, spin echo (SE): TR, 600 ms; TE, 15 ms; slice thickness, 2 mm; reconstructed pixel size, 0.48 × 0.48 mm; 16 slices.
- Pre- and post-contrast T1W, 2D, coronal, SE: TR, 675 ms; TE, 15 ms; slice thickness, 2 mm; reconstructed pixel size, 0.49 × 0.49 mm; 20 slices.

Images were analyzed using Medical Image Merge (MIM) software version 6.9 (Cleveland, OH, USA).

**Hematology and serum chemistry**. A complete blood cell count was performed on a Sysmex XT-2000iV hematology instrument (Sysmex America, Lincolnshire, IL, USA). Plasma and serum were obtained after incubation at room temperature for 10 min and subsequent centrifugation at 1800 x *g* for 10 min. Serum chemistry was analyzed on a Piccolo Xpress analyzer using the Piccolo general chemistry 13 panel (Abaxis, Parsipanny, NJ, USA).

**Cytology**. Samples of vitreous fluid from the eyes were collected via syringe and hypodermic needle, placed on uncharged slides, airdried on a hotplate for 10 min at 60°C, and then fixed in methanol. Slides were irradiated at 5 MRad using a JL Shepherd 484-R2 Co[60] irradiator. Routine Wright–Giemsa staining was performed with Epredia Shandon Kwik-Diff (cat# 99-907-00; Thermo Fisher Scientific, Waltham, MA, USA). In situ hybridization to detect EBOV nucleic acid (RNA) was performed using the manual RNAscope 2.5 HD RED kit (cat# 322360; Advanced Cell Diagnostics, Newark, CA, USA) according to the manufacturer's instructions, with procedural modifications for protocol optimization (including elimination of heat-induced target retrieval and reduction of time and temperature of enzymatic digestion in dilute proteolytic solution), validated by appropriate controls, using EBOV-VP40 (genomic) RNA probe (cat# 507141; Advanced Cell Diagnostics) and EBOV-VP35 (antigenomic) RNA probe (cat# 527491; Advanced Cell Diagnostics), with hematoxylin (blue) counterstain (cat# 7211; Thermo Fisher Scientific).

**Virological assays**. Plasma, cerebrospinal fluid, vaginal swab medium, aqueous and vitreous humor were inactivated in TRIzol LS according to the manufacturer's instructions (Thermo Fisher Scientific, Waltham, MA, USA) and removed from the BSL-4. RNA was isolated using QIAamp Viral RNA Mini Kit (Qiagen, Germantown, MD, USA) BEI Resources Critical Reagents Program (CRP) EZ1 RT-PCR kit assay according to the manufacturer's manual[32] and analyzed on an QuantStudio 7 Flex Real-Time PCR instrument (Thermo Fisher Scientific) with a lower limit of quantitation of $1 \times 10^2$ genome equivalents (GEq) per reaction. Attempts to isolate infectious EBOV from aqueous and vitreous humor were performed by making a 1:10 dilution of sample in Gibco minimum essential medium (MEM; Thermo Fisher Scientific) and adsorption of inoculum on Vero E6 cell monolayers (#NR-596; Vero C1008 [E6] grivet kidney cells, American Type Culture Collection [ATCC]) at 37°C for 1 h. Following inoculation, cell cultures were monitored for presence of cytopathic effect (CPE) for up to 7 d, and supernatant was harvested for RNA extraction followed by RT-qPCR analysis as described.

**Immunological assays**. Cells contained in the vitreous humor collected terminally on day 99 post-exposure from OS and OD were analyzed by flow cytometry. A list of antibodies used for staining of cells in vitreous humor is provided in Supplementary Table 7. Briefly, fresh vitreous humor was blocked with 5 µL of Human TruStain Fc Receptor Blocking Solution (cat#422302; BioLegend, San Diego, CA, USABiolegend) on ice for 10 min followed by staining with a 100 µL master mix of antibodies in phosphate-buffered saline (PBS) on ice for 20–30 min. A total of 50 µL of count bright beads (cat# C36950, lot# 2014179; Thermo Fisher Scientific) were added to each sample for downstream enumeration of cell numbers. Red blood cells were lysed with 1 mL of 1x BD FACS Lyse (BD Biosciences, Franklin Lakes, NJ, USA) per tube and incubated for 10 min at room temperature. Surface-stained cells were washed with 3 mL of PBS-2% FBS-2mM EDTA (PBS-2), spun at 500 x *g* for 5 min; the supernatant was discarded and 500 µL of Cytofix/Cytoperm was added, followed by incubation for 30 min at room temperature. Fixed and inactivated cells were washed and centrifuged as previously described, followed by resuspension in 350 µL of PBS for flow cytometry acquisition on a LSRII Fortessa instrument (BD Biosciences).

FlowJo software version 10 (FlowJo, Ashland, OR, USA) was used for data analysis. The gating strategy was as follows: FSC-A/FSC-H and SSC-W/SSC-A gating was used to identify singlets. Viability dye was used to exclude dead cells.

Live cells were gated for CD45 positivity and plotted on FSC-A/SSC-A to identify CD45+ cells of the appropriate size. CD3+ cells were further divided into CD8+ T-cells or CD4+ T-cells. CD4+ and CD8+ T-cells were further identified as naïve (CD28+, CD95−), central memory (CD28+, CD95+), or effector memory (CD28−, CD95+) populations. Of the live CD45+ cells, B-cells were not observed but characterized as naïve (CD20+, CD3−, CD27−, IgD+), marginal-zone (CD20+, CD3−, CD27+, IgD+), or memory B-cells (CD20+, CD3−, CD27+, IgD−). Cells within the CD45+/CD20−/CD3− gate were divided into monocytes (CD14+), plasmacytoid dendritic cells (CD14−, HLA-DR+, CD123+), or myeloid dendritic cells (CD14−, HLA-DR+, CD123−, CD11c+).

Proinflammatory cytokines were measured in the vitreous humor of OS and OD of the infected animal and in the vitreous humor of an uninfected control animal using the MILLIPLEX MAP Non-Human Primate Cytokine Magnetic Bead Panel (PCYTMG-40K-PX23; Millipore Sigma, Burlington, MA, USA) on a Luminex FLEXMAP 3D instrument (Luminex Corporation, Austin, TX, USA) according to the manufacturer's recommendations (Supplementary Table 8).

Anti-EBOV glycoprotein (GP$_{1,2}$) immunoglobulin G (IgG) was measured in plasma and vitreous humor of OS and OD[33]. Additionally, rhesus IgG were measured in the vitreous humor of OS and OD using a rhesus monkey immunoglobulin G ELISA kit (Molecular Innovations, Novi, MI, USA) according to the manufacturer's instructions for calculation of the Goldmann–Witmer Coefficient (GWC). A GWC of ≥ 4 is diagnostic of a local antibody production to a specific pathogen and was calculated using formula 1 and 2.

$$GWC = \frac{EBOV\ IgG_{vitreous}/Total\ IgG_{vitreous}}{EBOV\ IgG_{serum}/Total\ IgG_{serum}} \quad (1)$$

$$GWC = \frac{174739.61\ EU \cdot mL^{-1}/14536864.29\ ng \cdot mL^{-1}}{20682.84\ EU \cdot mL^{-1}/9974181.106\ ng \cdot mL^{-1}} = 5.7968 \quad (2)$$

**Histopathology**. The eyes were fixed in 10% neutral buffered formalin (NBF) for 72 h, followed by 12 h post-fixation in Davidson's solution. Both eyes were sectioned in an anterior–posterior plane, routinely processed, paraffin-embedded, sectioned at 4 µm via microtome, and mounted on charged glass slides. Routine hematoxylin and eosin (H&E) staining was performed using CAT Modified Lillie-Mayer's Hematoxylin (cat# CATHE-GL; Biocare Medical, Pacheco, CA, USA) and Epredia Richard-Allan Scientific Eosin-Y with Phloxine (cat# 22-050-198; Thermo Fisher Scientific, Waltham, MA, USA), and slides were evaluated microscopically with a Leica DM3000 microscope (Leica Microsystems, Buffalo Grove, IL). Images were captured with an Olympus DP74 camera and cellSens Standard 3.1 software (Olympus America Inc., Waltham, MA, USA).

Immunohistochemistry was performed using mouse anti-EBOV matrix protein (VP40 [3G5]) antibody at 1:3000 (cat# 0201-016; IBT Bioservices, Rockville, MD, USA) or rabbit anti-EBOV glycoprotein (GP$_{1,2}$) antibody at 1:3000 (cat# 0301-015; IBT Bioservices) as previously described[34]. Stained slides were visualized with Warp red chromagen (cat# WR806S; Biocare Medical, Pacheco, CA, USA) and hematoxylin (blue) counterstain (cat# 7211; Richard-Allan Scientific, Thermo Fisher Scientific, Waltham, MA, USA). Additional IHC staining was performed using mouse anti-CD4 [BC/1F6] at 1:80 (cat# CM153B; Biocare Medical, Pacheco, CA, USA), rabbit anti-CD8 at 1:1500 (cat# CM154; Biocare Medical), rabbit anti-CD38 at 1:650 (cat# LS-A9696; LS-Bio, Seattle, WA, USA), and mouse anti-CD68 at 1:500 (cat# NBP-74570; Novus Biologicals, Centennial, CO, USA). In situ hybridization (ISH) was performed using the RNAscope 2.5 HD RED reagent kit (cat# 322360; Advanced Cell Diagnostics), EBOV-VP40 (genomic) RNA probe (cat# 507141; Advanced Cell Diagnostics), and EBOV-VP35 (antigenomic or replicative intermediate) RNA probe (cat# 527491; Advanced Cell Diagnostics) to detect EBOV RNA (Advanced Cell Diagnostics, Newark, CA, USA) visualized with hematoxylin (blue) counterstain (cat# 7211; Thermo Fisher Scientific)[35]. Histochemistry (special staining) was performed using a Trichrome, Masson, Aniline Blue Stain Kit (cat# 9179 A; Newcomer Supply, Middleton, WI, USA; Supplementary Table 9).

**Data analysis**. BD FACS Diva software version 6.1.3 (BD Biosciences, Franklin Lakes, NJ, USA) was used for collection of raw flow cytometry data. FlowJo software version 10 (FlowJo, Ashland, OR, USA) was used for analysis of flow cytometry data. MIM software version 6.9 (Cleveland, OH, USA) was used for analysis of MR images. Initial graphs were generated using GraphPad software version 8.4.2 (Prism, La Jolla, CA, USA) and final artwork was created in Adobe Illustrator 25.4.8. (Adobe, San Jose, CA, USA; Supplementary Table 9).

**Reporting summary**. Further information on research design is available in the Nature Research Reporting Summary linked to this article.

## Data availability

Data generated or analyzed during this study are included in this article and its supplementary files. All other source data are available from the corresponding authors on reasonable request.

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

## Acknowledgements

We thank the staff of the National Institutes of Health (NIH) National Institute of Allergy and Infectious Diseases (NIAID) Division of Clinical Research (DCR) Integrated Research Facility at Fort Detrick (IRF-Frederick) who supported this study, in particular Joseph Laux and Irwin Feuerstein. We thank Dima Hammoud (NIH Clinical Center) for interpretation of magnetic resonance scans. We are grateful to Alcides Fernandes Filho (Emory Eye Center) for generously providing an iCare device. We thank Jiro Wada (IRF-Frederick) for assisting in graphical artwork design and Anya Crane (IRF-Frederick) for text editing. This research was supported in part through the National Institutes of Health (NIH) National Institute of Allergy and Infectious Diseases (NIAID) prime contract with Battelle Memorial Institute (Contract No. HHSN272200700016I) and subsequently with Laulima Government Solutions (Contract No. HHSN272201800013C). G.W., T.K.C., A.M.W.H., S.E.K., R.B., J.R.K., S.M.A., N.M.A., D.R., J.H.L., M.R.H., and J.H.K. performed this work as employees under these contracts. J.H.L. and J.H.K. also performed this work as employees of Tunnell Government Services, a subcontractor of Battelle Memorial Institute and Laulima Government Solutions. This project was funded in part by NIH grant No. U19AI142790 (E.O.S.). This project was funded in part with federal funds from the NIH National Cancer Institute (NCI), under Contract No. HHSN261201500003I, Task Order No. HHSN26100043, and Contract No. 75N91019D00024, Task Order No. 75N91019F00130 (I.C.). The funders had no role in the design of the study; in the collection, analyses, or interpretation of data; in the writing of the manuscript, or in the decision to publish the results. The content of this publication does not necessarily reflect the views or policies of the U.S. Department of Health and Human Services (HHS) or of the institutions and companies affiliated with the authors. Mention of trade names, commercial products, or organizations does not imply endorsement by the U.S. Government. The experiments described in this manuscript were approved by the United States of America Department of Health and Human Services (HHS), National Institutes of Health (NIH), Division of Clinical Research (DCR) Integrated Research Facility at Fort Detrick (IRF-Frederick), Frederick, MD, USA Animal Care and Use Committee in compliance with all applicable federal regulations governing the protection of animals and research.

## Author contributions

Conceptualization: G.W., C.D., R.A., E.O.S., and I.C. Methodology: G.W., T.K.C., A.M.W.H., R.B., J.R.K., N.M.A., D.R., and I.C. Investigation: G.W., T.K.C., S.Y., J.G.S., S.E.K., M.S.C., E.O.S., and I.C. Visualization: G.W., T.K.C., A.M.W.H., S.E.K., J.H.L., J.R.K., S.M.A., and N.M.A. Funding acquisition: E.O.S. Project administration: G.W. Supervision: G.W., M.R.H., and J.H.K. Writing – original draft: G.W., J.H.K., and I.C. Writing – review & editing: G.W., T.K.C., S.Y., J.G.S., A.M.W.H., S.E.K., R.B., J.R.K., S.M.A., N.M.A., D.R., J.H.L., M.S.C., C.D., R.A., M.R.H., J.H.K., E.O.S., and I.C.

## Funding

## Competing interests

The authors declare no competing interests
