## [Peer Review File · Communications Biology]

Reviewers' comments:

Reviewer #1 (Remarks to the Author):

In this manuscript "View on Ebola virus: The Eyes Have It" Gabriella Worwa and colleagues unveil their data on a single monkey with what the authors refer to as persistent EBOV sequela. The authors performed unbelievable amounts of research on this single monkey to tease out the unilateral uveitis pathology associated with surviving EBOV infection. The weakest points of this manuscript include a single specimen and no data to show persistent EBOV infection. However, the research is detailed and well-performed and should add some value to the general audience of the journal. Below, I made a few comments, which must be addressed, to enhance scientific readability of the manuscript:

- 1) The title must be changed and must state the actual findings
- 2) A few of subtitles such as line 87 "Of man and macaque: questions for the animal model" must be changed to fit scientific journals.
- 3) The authors state "Also, we confirm a causative role of persistent EBOV in the pathogenesis of EVD-associated uveitis via the direct detection of EBOV RNA and indirect detection of intraocular EBOV-specific antibodies." I do not find their findings compelling enough to show cause and effect relationship. The authors need to dial back their enthusiasm a bit. Just because they found EBOV RNA this by itself does not mean EBOV RNA was the cause of the disease.
- 4) In the discussion the authors repeated the same points many times. The discussion should be trimmed by at least 30%. The discussion should include the points that the reviewer suggests-see below.
- 5) My understanding is that panuveitis and EBOV persistence is rare in NHPs following IM injection of the virus. The infection route during an outbreak is most likely not due to IM exposure. The majority of the infections happen through the mucosal membranes including eye, respiratory, and mouth. I do not believe the IM route of infection faithfully represents pathophysiological findings which resembles the actual endemic exposure. The authors need to spend a few sentences in the discussion on this specific area.
- 6) Although the authors attempted to directly connect panuveitis and EBOV persistence to the infection only, there is a disconnect between Ebola infection and health of the NHP at the beginning of the study. The authors need to expand on this caveat and put a few sentences in the discussion regarding how pre-existing conditions may have directed the virus to the eye.
- 7) In the study which this monkey was chosen, the infection route was via IM and antibody treatment started at day 4. The authors must discuss the implications of the antibody treatment and the timing to their findings.
- 8) The study used a single animal but the manuscript has 20 or so authors. I am having difficult time to understand how these 20 authors contributed to a study that used only one monkey.

Reviewer #2 (Remarks to the Author):

The authors analyzed experimental Ebola virus infection in a rhesus monkey survivor and characterized the clinical, virologic, immunologic, and histopathologic features of uveitis. They also identified a persistent Ebola virus as a driver of severe chronic inflammation.

I suggest corrections:

1. In discussion, please state if there could be any errors due to a human factor in measuring the size of a globe on MRI. In humans, the size of an eyeball does not vary so much with changes in the intraocular pressure.
2. In discussion, on page 12, lines 257-259, iris heterochromia can be not only due to pigment cell involvement but also due to iris stroma atrophy, so please take this into account as well.

Reviewer #3 (Remarks to the Author):

The manuscript by Worwa et al., is a detailed case report of uveitis that developed in a rhesus macaque following infection with EBOV and treatment with a monoclonal antibody. Uveitis has emerged as a major complication for Ebola virus disease (EVD) survivors, most notably following the longest and largest EBOV epidemic to date that took place between December 2013 and June 2016 in west Africa. This a careful and detailed study of a unique case and provides valuable insight into this complication that has not been attainable from clinical studies. Some minor revisions are requested to improve accessibility of the manuscript and interpretation of the findings reported:

1. The introduction is constructed in a highly unusual manner that renders it rather unwieldy. Strongly recommend that the authors streamline the introduction, remove the sub-headings and turn it into a concise summary of our current understanding of EVD related uveitis
2. Replace the word "exposure", "exposed" with infection and infected when describing experimental inoculation of the macaque with EBOV
3. Add information pertaining to the challenge and monoclonal antibody treatment to the timeline in Figure 1
4. Provide viral loads in the blood, antibody titers, and other clinical information pertinent to EBOV challenge for this animal – this is critical for understanding disease presentation and status of the animal
5. Under virologic and immunologic assessment section, please clarify the sampling of aqueous versus vitreous humor in the eye. As currently stated, it looks like aqueous humor was collected days 30, 37, 58 and 79 while vitreous humor was only collected day 99. Are these terms being used interchangeably?
6. Please provide raw data for measuring antibodies in the vitreous humor – OD in this site versus plasma at the same time point for instance
7. Cytology suggests that the majority of CD45+ cells in vitreous humor are T cells; however, IHC staining indicate a large presence of plasma B cells. How do the authors explain this difference?
8. The discussion repeats much of the results – strongly suggest that it be condensed and focused on data interpretation and discussion instead. For instance, do the authors think the T cells in the vitreous humor are antigen-specific?

Response to Reviewers' comments

Reviewer #1 (Remarks to the Author):

In this manuscript “View on Ebola virus: The Eyes Have It” Gabriella Worwa and colleagues unveil their data on a single monkey with what the authors refer to as persistent EBOV sequela. The authors performed unbelievable amounts of research on this single monkey to tease out the unilateral uveitis pathology associated with surviving EBOV infection. The weakest points of this manuscript include a single specimen and no data to show persistent EBOV infection. However, the research is detailed and well-performed and should add some value to the general audience of the journal.

Our response: We thank the reviewer for appreciating the amount and quality of work we performed.

Below, I made a few comments, which must be addressed, to enhance scientific readability of the manuscript:

1) The title must be changed and must state the actual findings.

Our response: We agree with the reviewer that the title was subideal. Toward neither under nor overstating our findings, we have replaced the title completely, being careful to only state the association in a newly proposed title such that “View on Ebola virus: The Eyes Have It” is now “Severe uveitis associated with persistent intraocular Ebola virus RNA in a convalescent rhesus monkey”.

2) A few of subtitles such as line 87 “Of man and macaque: questions for the animal model” must be changed to fit scientific journals.

Our response: We agree with the reviewer (and also Reviewer 3) that the Introduction should be shortened and clarified. As part of that effort, we have removed the subheadings entirely, sometimes adding framing language when necessary for context, and generally shortened/clarified the text. In the revised manuscript, the introductory flow is now framed as P1) introductory background to EVD-associated uveitis; shortened P2) outstanding questions/gaps that still need answers; P3) challenges associated with using the animal model to get to those answers; and P4) setup of the current report.

3) The authors state “Also, we confirm a causative role of persistent EBOV in the pathogenesis of EVD-associated uveitis via the direct detection of EBOV RNA and indirect detection of intraocular EBOV-specific antibodies.” I do not find their findings compelling enough to show cause and effect relationship. The authors need to dial back their enthusiasm a bit. Just because they found EBOV RNA this by itself does not mean EBOV RNA was the cause of the disease.

Our response: We agree with the reviewer that caution is needed to avoid over-interpretation of results, especially in a single observation. We agree that our initially submitted wording may have implied causation and have hopefully modified the message appropriately in the revised manuscript. While the timing of uveitis onset, detection of intraocular EBOV RNA, and a positive Goldmann-Witmer coefficient potentially implicate causation, we have tried to revise language away from “causation” to “association” of EBOV RNA and EBOV-specific IgG with the clinical syndrome and the gross/histopathology throughout. In fact, we have proactively introduced language cautioning against

making inappropriate causal conclusions. Also, in response to broader reviewer comments, we introduced additional language covering the possibility that we cannot rule out pre-existing coinfection or other pre-dispositions. Some examples of the revised language in relevant manuscript sections include:

- Revised abstract: “The association of persistent EBOV RNA as potential driver of severe immunopathology has pathophysiologic implications for understanding ... ”;
- Introduction paragraph 4: “Causality should not be assumed, but the association may implicate persistent EBOV RNA as potential driver of severe ocular immunopathology and inform future efforts toward understanding, preventing, and treating vision-threatening uveitis in EVD survivors”;
- Discussion paragraph 2: “Care is needed to avoid over-interpreting these data or assuming causal relationships from a single observation. However, in the context of vision-threatening intraocular pathology (diffuse uveal lymphoplasmacytic infiltration with fibrosing cyclitic membrane), the detection of high levels of intravitreal EBOV-specific IgG in tissue proximity to detectable EBOV RNA at least query a role for persistent EBOV RNA as potential driver of uveitis and its complications”;
- Discussion final paragraph: “In this rhesus monkey survivor of experimental EBOV infection, we provide a first report of the detailed clinical features, evolution, and the severe immunopathologic consequences of EVD-associated uveitis. We report intraocular EBOV-specific antibody detection and persistent intraocular EBOV RNA months after clearance of viremia and even after clinical resolution. Causality should be not assumed, especially in the context of a single observation, but the association of persistent EBOV RNA as potential driver of severe immunopathology has pathophysiologic implications for understanding, preventing, and treating vision-threatening uveitis in Ebola virus disease survivors”;
- Discussion paragraph 4: We added language in the Discussion re. other factors that may have predisposed this NHP to develop uveitis (this directly addresses your comment #6): “The role of pre-existing host genetic, host immune, and bacterial, fungal or viral coinfections in the development of uveitis associated with EVD is not clear, including for this rhesus monkey. Indeed, without baseline assessment, it is possible that subclinical uveitis may have been present prior to EBOV infection and activated during and in the course of the disease”;
- We also adjusted the Figure 2 title language in this regard.

4) In the discussion the authors repeated the same points many times. The discussion should be trimmed by at least 30%. The discussion should include the points that the reviewer suggests-see below.

Our response: We agree with the reviewer (and also Reviewer #3) that we should have been less verbose in the Discussion (as well as with comments to streamline and clarify the Introduction as pointed out by Reviewer #3). We have revised the manuscript to a) avoid and minimize redundancy and b) maintain a clear/connected Discussion narrative. Briefly, we have removed redundant statements and placed more emphasis on discussing our findings and connecting those findings to their implications. Rather than

summarizing them here, please read through the revised Discussion section, but note here the general flow of the Discussion outlined as:

- P1: summary of the NHP findings (given the complexity of the results section, we feel it is still important to summarize in a topline manner to set the stage for the rest of the Discussion);
- P2: Zoom out for contextual interpretation of the NHP findings (the host-pathogen interaction in the environment of ocular immune privilege);
- P3: Parallel comparison of the NHP and human “equivalent” (both unusual n of 1 observations);
- P4: Implications of the findings for understanding risk in humans;
- P5: Implications for pathogenesis: understanding tissue-specific location and the vitreous interface;
- P6: Implication for pathogenesis: ocular tropism and cellular involvement in the “itis” and its complications;
- P7: Implications for clinical management in EVD survivors; and
- P8: Summary and conclusion.

5) My understanding is that panuveitis and EBOV persistence is rare in NHPs following IM injection of the virus. The infection route during an outbreak is most likely not due to IM exposure. The majority of the infections happen through the mucosal membranes including eye, respiratory, and mouth. I do not believe the IM route of infection faithfully represents pathophysiological findings which resembles the actual endemic exposure. The authors need to spend a few sentences in the discussion on this specific area.

Our response: We agree with the reviewer. Indeed, panuveitis (clinically) associated with EBOV has never been described in NHPs, though this is very likely related to the great difficulty in doing these kinds of detailed studies prospectively when a) “true” survivors are rare, and, even after a therapeutic, b) following NHPs for months after acute disease is a logistic and financial challenge. Detection of viral RNA in the eye this far out has also never been described in an NHP or any other animal model (hence the parallel comparison to the human case that is still the only example of the EBOV RNA detected in the eye). The only prior finding has been the retrospective detection of EBOV in the eyes of “delayed death” NHPs in archived tissues (reference 18 cited in the manuscript) but this detection occurred during the acute period. We strongly agree with the reviewer’s comment regarding the 1000-pfu IM route NOT reflecting “natural” human exposure. However, aside from direct conjunctival exposure to EBOV, all other routes of exposure (e.g., IM, SQ, aerosol, oral). share some commonalities in developing widespread viremia. We speculate that “breach” into the eye occurred during the secondary viremia phase. If true, the original route of exposure, with exception of conjunctival exposure, is less relevant as long as viremia occurs. We follow and understand the reviewer’s point, but from our viewpoint the NHP model does reproduce EBOV viremia and is generally accepted as reproducing the overall genetic and anatomical similarities among NHPs and humans to enable valuable comparisons and to draw lessons that can be learned.

To accommodate the reviewer’s concern, we added the following text to the Discussion paragraph on risk factors:

“Most human infections occur at mucosal interfaces rather than the intramuscular inoculation in our experimental subject; the impact of the route of initial EBOV infection on the subsequent development of

uveitis in human EVD survivors is unclear. In humans and experimentally infected monkeys, however, with few exceptions, the common theme is systemic viremia that affords, especially when at high levels and extended duration, breach of the blood-retinal and blood-aqueous barriers to seed the eye”.

6) Although the authors attempted to directly connect panuveitis and EBOV persistence to the infection only, there is a disconnect between Ebola infection and health of the NHP at the beginning of the study. The authors need to expand on this caveat and put a few sentences in the discussion regarding how pre-existing conditions may have directed the virus to the eye.

Our response: We agree with the reviewer that this is a limitation we failed to address. To put our findings into a more balanced perspective, we added the following sentences to the Discussion paragraph on risk factors:

“The role of pre-existing host genetic, host immune, and bacterial, fungal or viral coinfections in the development of uveitis associated with EVD is not clear, including for this rhesus monkey. Indeed, without baseline assessment, it is possible that subclinical uveitis may have been present prior to EBOV infection and activated during and in the course of the disease.”

7) In the study which this monkey was chosen, the infection route was via IM and antibody treatment started at day 4. The authors must discuss the implications of the antibody treatment and the timing to their findings.

Our response: We agree with the reviewer that the potential relationship between the acute EVD illness and subsequent uveitis logically requires more detail **and have added new data collected from this NHP during the acute phase of EBOV infection to present a more complete clinical picture. The data were added to Figure 1 and 2**, and include the following items:

- Complete EBOV RNA load determined from all available time points before and after EBOV inoculation analyzed by RT-qPCR;
- Endogenous humoral immune response (rhesus anti-EBOV GP IgG titers in serum detected by ELISA over time);
- Euthanasia scores based on clinical cage-side observations of this NHP to show the degree of clinical signs presented by this NHP during the acute phase;
- Concentrations of aspartate aminotransferase (AST) to Figure 1 as relevant biomarker of acute EBOV infection;
- Platelets numbers as relevant readout of acute EBOV infection;
- Exact timing of EBOV inoculation and therapeutic antibody administration in Figure 1; and
- Flow cytometry data obtained from vitreous humor collected terminally showing B and T cell populations identified.

We also agree that the Discussion should include language exploring the potential relationship between the antibody therapeutic and uveitis, including possible “generic” effects (relationship to survival and extended viremia) and (unknown) effect specific to the class or this actual therapeutic. We have added language in this regard to Discussion paragraph 4:

“It is clear that receipt of an EBOV-specific therapeutic was generically related to the NHP’s survival (and perhaps to extended duration of a biphasic viremia); what is not clear is any specific

relationship between the development of uveitis and receipt of a mAb-based therapeutic in general, or this mAb in particular.”

8) The study used a single animal but the manuscript has 20 or so authors. I am having difficult time to understand how these 20 authors contributed to a study that used only one monkey.

Our response: We appreciate the reviewer’s curiosity, but we would like to assure reviewers and the editor that every co-author participated actively in the clinical follow-up and contributed to the resulting manuscript. Indeed we spell out the reasons why these data are so challenging to obtain in the Introduction paragraph 3: although only one animal was used, the nature of procedures performed on this NHP were not only unique but also presented logistical challenges due to the required biosafety level 4 environment. For example, the ophthalmic assessments, the anterior chamber paracentesis, and the medical imaging alone required several teams of specialized staff as such analyses have never been performed before. Similarly, the list of analyses performed using samples collected from this NHP was long and diverse and required different teams/experts to generate the results obtained. Eventually, it was our opinion that the amount of time and resources spent on this animal, and the interesting results obtained, warranted submission of our findings for publication and inclusion of all involved.

Reviewer #2 (Remarks to the Author):

The authors analyzed experimental Ebola virus infection in a rhesus monkey survivor and characterized the clinical, virologic, immunologic, and histopathologic features of uveitis. They also identified a persistent Ebola virus as a driver of severe chronic inflammation.

I suggest corrections:

1) In discussion, please state if there could be any errors due to a human factor in measuring the size of a globe on MRI. In humans, the size of an eyeball does not vary so much with changes in the intraocular pressure.

Our response: We thank the reviewer for raising this point. As with any measurement, it is possible to have human error. However, the way we reduced the extent of imprecision was by using the same plane of each MR image and ensuring that all anatomical landmarks lined up. The contour of the eye was then encircled, and the volume determined computationally. We trust that the smaller volumes estimated in the affected eye are correct, especially in relation to the right eye. To address the reviewer’s concern, we have added the following language in the appropriate Results and Discussion sections:

- Results: “ Because measurement of intraocular pressure using applanation tonometry proved challenging in the ABSL-4 negative-pressure environment, we investigated the volumetric size of the globes by MRI as a structural approximation of decreased intraocular pressure. Recognizing that determining ocular globe volume from MR images is susceptible to measurement error, we utilized the same MRI plane with all anatomical landmarks lined up, encircled the globe, and determined the volume computationally“; and

- Discussion: “Though measurement errors are possible when extrapolating the globe volumes from MR images, hypotony was presumably present in the left eye when compared to the right eye”.

2) In discussion, on page 12, lines 257-259, iris heterochromia can be not only due to pigment cell involvement but also due to iris stroma atrophy, so please take this into account as well.

Our response: We agree with the reviewer. We modified our sentence, which now states “Finally, iris heterochromia observed in the human survivor with panuveitis and EBOV persistence 13,14 is of uncertain pathogenesis but either the direct involvement of pigmented iris epithelial cells, and/or iris stroma atrophy indirectly may be implicated.”

Reviewer #3 (Remarks to the Author):

The manuscript by Worwa et al., is a detailed case report of uveitis that developed in a rhesus macaque following infection with EBOV and treatment with a monoclonal antibody. Uveitis has emerged as a major complication for Ebola virus disease (EVD) survivors, most notably following the longest and largest EBOV epidemic to date that took place between December 2013 and June 2016 in west Africa. This a careful and detailed study of a unique case and provides valuable insight into this complication that has not been attainable from clinical studies.

Our response: We thank the reviewer for the positive assessment of our work.

Some minor revisions are requested to improve accessibility of the manuscript and interpretation of the findings reported:

1) The introduction is constructed in a highly unusual manner that renders it rather unwieldy. Strongly recommend that the authors streamline the introduction, remove the sub-headings and turn it into a concise summary of our current understanding of EVD related uveitis

Our response: We agree with the reviewer (and also Reviewer 1) that the Introduction should be shortened and clarified. As part of that effort, we have removed the subheadings entirely, sometimes adding framing language when necessary for context, and generally shortened/clarified the text. In the revised manuscript, the introductory flow is now framed as P1) introductory background to EVD-associated uveitis; shortened P2) outstanding questions/gaps that still need answers; P3) challenges associated with using the animal model to get to those answers; and P4) setup of the current report.

2) Replace the word “exposure”, “exposed” with infection and infected when describing experimental inoculation of the macaque with EBOV.

Our response: Done (7 times).

3) Add information pertaining to the challenge and monoclonal antibody treatment to the timeline in Figure 1.

Our response: We agree with the reviewer that this information will be useful to the reader. We included the exposure and treatment information in our revised Figure 1 as suggested. Additionally, a citation (Milligan *et al.*, 2022) was added that describes the antibody used for treatment.

4) Provide viral loads in the blood, antibody titers, and other clinical information pertinent to EBOV challenge for this animal – this is critical for understanding disease presentation and status of the animal.

Our response: We agree with the reviewer that the potential relationship between the acute EVD illness and subsequent uveitis logically requires more detail **and have added new data collected from this NHP during the acute phase of EBOV infection to present a more complete clinical picture. The data were added to Figure 1 and 2**, and include the following items:

- Complete EBOV RNA load determined from all available time points before and after EBOV inoculation analyzed by RT-qPCR;
- Endogenous humoral immune response (rhesus anti-EBOV GP IgG titers in serum detected by ELISA over time);
- Euthanasia scores based on clinical cage-side observations of this NHP to show the degree of clinical signs presented by this NHP during the acute phase;
- Concentrations of aspartate aminotransferase (AST) to Figure 1 as relevant biomarker of acute EBOV infection;
- Platelets numbers as relevant readout of acute EBOV infection;
- Exact timing of EBOV inoculation and therapeutic antibody administration in Figure 1; and

Flow cytometry data obtained from vitreous humor collected terminally showing B and T cell populations identified.

5) Under virologic and immunologic assessment section, please clarify the sampling of aqueous versus vitreous humor in the eye. As currently stated, it looks like aqueous humor was collected days 30, 37, 58 and 79 while vitreous humor was only collected day 99. Are these terms being used interchangeably?

Our response: The reviewer raises an important question: the terms are not used interchangeably. Aspiration of aqueous humor from the anterior chamber was logistically difficult (due to BSL-4 biocontainment) but feasible with anterior chamber paracentesis and we were able to aseptically collect small volumes of aqueous humor on days 30, 37, 58 and 79. Collection of vitreous humor from the posterior chamber is much more challenging and requires a surgical setting not available in BSL-4 biocontainment: careful collection of vitreous humor therefore could only be performed during terminal procedures.

6) Please provide raw data for measuring antibodies in the vitreous humor – OD in this site versus plasma at the same time point for instance.

Our response: On Day 99, the anti-EBOV GP IgG titer in serum was 4.21 log₁₀ whereas it was 5.24 log₁₀ in the vitreous humor of the left eye. In contrast, the vitreous humor of the right (normal) eye did not contain detectable anti-EBOV GP IgG antibodies as determined by ELISA (< OOR). For space reasons

we have not included this in the text thus far, but would of course happily do so if the editor advised us accordingly.

Day 99 Serum	Day 99 Vitreous humor, Left eye	Day 99 Vitreous humor, Right eye
16574.5	174739.6	< OOR

7) Cytology suggests that the majority of CD45+ cells in vitreous humor are T cells; however, IHC staining indicate a large presence of plasma B cells. How do the authors explain this difference?

Our response: We were surprised by this observation as well but do not doubt the findings. It is possible that the different samples collected contained different cell populations. Another explanation for this observation could be the limitations of the respective assays, perhaps the specificity and sensitivity of antibodies used in the IHC staining protocol.

The most sensitive and comprehensive method for detection of different cell populations is flow cytometry. To help address your question or concern, **we have added in Figure 2 a flow cytometry panel that shows the levels of the different B and T cell populations in vitreous humor.**

8) The discussion repeats much of the results – strongly suggest that it be condensed and focused on data interpretation and discussion instead. For instance, do the authors think the T cells in the vitreous humor are antigen-specific?

We agree with the reviewer (and also Reviewer #1) that we should have been less verbose in the Discussion. We have revised the manuscript to a) avoid and minimize redundancy and b) maintain a clear/connected Discussion narrative. Briefly, we have removed redundant statements and placed more emphasis on discussing our findings and connecting those findings to their implications. Rather than summarizing them here, please read through the revised Discussion section, but note here the general flow of the Discussion outlined as:

- P1: summary of the NHP findings (given the complexity of the results section, we feel it is still important to summarize in a topline manner to set the stage for the rest of the Discussion);
- P2: Zoom out for contextual interpretation of the NHP findings (the host-pathogen interaction in the environment of ocular immune privilege);
- P3: Parallel comparison of the NHP and human “equivalent” (both unusual n of 1 observations);
- P4: Implications of the findings for understanding risk in humans;
- P5: Implications for pathogenesis: understanding tissue-specific location and the vitreous interface;
- P6: Implication for pathogenesis: ocular tropism and cellular involvement in the “itis” and its complications;
- P7: Implications for clinical management in EVD survivors; and
- P8: Summary and conclusion.

With regards to the specific questions about intravitreal T-cells, we could not determine EBOV antigen-specificity, but agree it was an oversight not to comment on this. We have added the following text in paragraph 2 of the Discussion:

“We were unable to confirm the antigen-specificity of the T-cell intravitreal inflammatory infiltrate, but it’s plausible that an EBOV-antigen specific ocular immune response, especially after the onset of inflammation (the collapse of privilege), would also involve T-lymphocytes.”

REVIEWERS' COMMENTS:

Reviewer #1 (Remarks to the Author):

The authors answered all of my comments in a satisfactory manner. I have no additional comments.

Reviewer #3 (Remarks to the Author):

In this revised manuscript, the authors adequately addressed the concerns raised during the first submission. This is a much more cohesive and informative paper that provides new insight into EVD-associated uveitis and will be an asset to the field